# Precision Medicine in Metastatic Colorectal Cancer: Targeting ERBB2 (HER-2) Oncogene

**DOI:** 10.3390/cancers14153718

**Published:** 2022-07-30

**Authors:** Javier Torres-Jiménez, Jorge Esteban-Villarrubia, Reyes Ferreiro-Monteagudo

**Affiliations:** 1Medical Oncology Department, MD Anderson Cancer Center Madrid, 28033 Madrid, Spain; 2Medical Oncology Department, University Hospital Ramon y Cajal, 28034 Madrid, Spain; jorge.esteban@salud.madrid.org (J.E.-V.); mariareyes.ferreiro@salud.madrid.org (R.F.-M.); 3Medical Oncology Department, University Hospital 12 de Octubre, 28041 Madrid, Spain

**Keywords:** colorectal cancer, precision medicine, ERBB2, HER-2

## Abstract

**Simple Summary:**

Colorectal cancer (CRC) is the third most common cancer in terms of incidence rate in adults and the second most common cause of cancer-related death in Europe. The treatment of metastatic CRC (mCRC) is based on the use of chemotherapy, anti-vascular endothelial growth factor (VEGF), and anti-epidermal growth factor receptor (EGFR) for RAS wild-type tumors. Precision medicine tries to identify molecular alterations that could be treated with targeted therapies. Although ERBB2 (also known as HER-2) has an important therapeutic role in breast and esophagogastric cancer, there are no approved ERBB2-targeted therapies for mCRC. The purpose of this review is to describe the landscape of ERBB2-positive mCRC.

**Abstract:**

Colorectal cancer (CRC) is the third most common cancer in terms of incidence rate in adults and the second most common cause of cancer-related death in Europe. The treatment of metastatic CRC (mCRC) is based on the use of chemotherapy, anti-vascular endothelial growth factor (VEGF), and anti-epidermal growth factor receptor (EGFR) for RAS wild-type tumors. Precision medicine tries to identify molecular alterations that could be treated with targeted therapies. ERBB2 amplification (also known as HER-2) has been identified in 2–3% of patients with mCRC, but there are currently no approved ERBB2-targeted therapies for mCRC. The purpose of this review is to describe the molecular structure of ERBB2, clinical features of these patients, diagnosis of ERBB2 alterations, and the most relevant clinical trials with ERBB2-targeted therapies in mCRC.

## 1. Introduction

Colorectal cancer (CRC) is the third cancer in terms of incidence rate in adults and the second most common cause of cancer-related deaths in Europe [1,2,3]. A total of 25% of CRC patients have metastatic lesions at diagnosis, and almost 50% of patients with early-stage CRC will develop disseminated or metastatic disease. The median overall survival (mOS) for patients with metastatic colorectal cancer (mCRC) is approximately 30 months (m) with current standard-of-care-therapies, according to phase III clinical trials and real-world data [4].

Most patients with mCRC have incurable disease, and treatment is based on systemic therapy with palliative intent. Different combinations of chemotherapeutic agents ((5-fluorouracil 40 (5-FU)/leucovorin (LV)/oxaliplatin (FOLFOX), 5-FU/LV/irinotecan (FOLFIRI), 5- 41 FU/LV/oxaliplatin/irinotecan (FOLFOXIRI)) and anti-vascular endothelial growth factor (VEGF), such as bevacizumab and aflibercept, have been developed recently, and they are used in the first- and second-line of treatment of mCRC [5,6]. Other therapies (trifluridine/tipiracil, regorafenib, raltitrexed) are used in third-line and successive lines of treatment [7,8].

Precision medicine includes the integration of molecular tumor profiles into clinical decision-making in cancer treatment. In other words, it consists in the identification of molecular targets, which would allow starting treatment with targeted therapies. Precision medicine is a challenge in oncology and it is changing the routine clinical practice [9,10].

EGFR (epidermal growth factor receptor, also known as ERBB1) is one of the first oncogenic targets in mCRC. KRAS and NRAS (RAS, rat sarcoma virus) mutations are associated with primary resistance to anti-EGFR therapies, so cetuximab and panitumumab are indicated only for RAS wild-type tumors [5,6,11].

Several target molecular biomarkers have changed the landscape of treatment of mCRC. These targeted therapies have demonstrated their effectiveness in clinical trials, obtaining the approval of the regulatory agencies: encorafenib and cetuximab in v-raf murine sarcoma viral oncogene homolog B1 (BRAF) V600E mutations, larotrectinib or entrectinib in neurotrophic tyrosine receptor kinase (NTRK) fusions, nivolumab/ipilimumab or pembrolizumab in deficient mismatch repair/microsatellite instability-high and high tumor mutation burden [12,13,14,15,16,17].

HER-2 (human epidermal growth factor receptor 2, also known as ERBB2) is a predictive biomarker that allows for the use of targeted therapies in breast and esophagogastric cancer in routine clinical practice [18,19,20,21]. ERBB2 activation by ERBB2 gene amplification or mutations is associated with anti-EGFR resistance in patients with mCRC [22,23]. ERBB2 is now under investigation for precision medicine in patients with mCRC [24,25,26,27]. Several clinical trials have evaluated the function of ERBB2-targeted therapies in patients with ERBB2-positive mCRC [28,29,30]. Although these clinical trials have promising results, ERBB2-targeted therapies have not been approved for patients with ERBB2-positive mCRC.

This review focuses on the knowledge of targeting ERBB2 oncogene in mCRC in the era of precision medicine: ERBB2 receptor biology, clinical features, and diagnosis of patients with ERBB2-positive mCRC and clinical trials that evaluated targeting therapies in patients with ERBB2-positive mCRC.

## 2. Molecular Biology of HER2 Receptor

ERBB2 is part of the family of epidermal growth factor receptors (ERBB). This family represents a group of receptor tyrosine kinases (RTK). The other members of this family are EGFR (ERBB1), HER3 (ERBB3), and HER4 (ERBB4). After binding with diverse ligands such as epidermal growth factor (EGF) or epiregulin (EREG), these receptors are able to heterodimerize, which leads to autophosphorylation. This allows the binding of diverse downstream signaling molecules, resulting in the activation of multiple pathways such as mitogen-activated protein kinase (MAPK), phosphoinositide 3-kinase (PI3K), Src pathways, and signal transducer and activator of transcription (STAT) transcription factors. ERBB2 is the only receptor capable of heterodimerizing with other ERBB receptors without binding any ligand and has an important role in the transphosphorylation of their dimerization partner [31].

The best-known pathogenic mechanisms involved in ERBB2 aberrant activation are overexpression of ERBB2 and activating mutations, both leading to constitutive activation of the receptor. Overexpression of ERBB2 in the membrane can lead to ERBB2 homodimerization and ligand-independent activation. In CRC, both mechanisms have been described. Traditionally, ERBB2 alterations have been considered to be mutually exclusive with KRAS/NRAS/BRAF alterations, although rare exceptions have been reported [32]. Modern series report that ERBB2 alterations are present in approximately 5% of CRC patients [32]. ERBB2 amplification would be present in approximately 3% of patients [33,34,35,36], while ERBB2-activating mutations in less than 2%. Co-existing amplifications and mutations would represent less than 1% of patients [32]. ERBB2-activating mutations are located in diverse regions of the receptor, such as extracellular domain II (S310F), juxtamembrane region (R678Q), and kinase domain (L775S, L866M, V777L, V842I, R868W). ERBB2 inhibition with Neratinib and Afatinib (two EGFR tyrosine kinase inhibitors) resulted in diminished cell growth in transfected cell lines [37]. Consistent with these findings, expression of ERBB2 by IHC (immunochemistry, membrane, and cytoplasmic staining) was significantly higher in adenomas compared to normal colorectal mucosa, and was significantly higher in adenocarcinomas compared to adenomas, suggesting a role in tumorigenesis [38].

Figure 1 represents an overview of HER2 signaling and different mechanisms of action of targeted therapies that will be further discussed in the article.

## 3. Diagnosis of HER2-Positive in mCRC

Reported rates of ERBB2 positivity have varied widely in earlier studies, due to differences in antibody clone selection, scoring criteria, staining platform, and cohort composition. Scoring criteria used in other carcinomas in which ERBB2 has a pathogenic role (breast and gastroesophageal) can produce false results, as some differences in ERBB2 expression have been noted. For example, ERBB2 expression in CRC cells is often restricted to the basolateral membranes of tumor cells and stains uniformly across the tumor. These patterns are different from breast (uniform staining across the membrane) and gastric patterns (basolateral staining but patchy pattern) [39]. These observations led to the development of a validated scoring system, the HERACLES, used in the HERACLES-A trial that is discussed later.

In the HERACLES diagnostic criteria, the pattern of expression, intensity of staining, and percentage of positive cells are used to define positivity. This is defined by intense (3+) expression in ≥50% of cells. Equivocal cases are defined by moderate (2+) expression in ≥50% or 3+ ERBB2 in more than 10% but less than 50% of tumor cells. These equivocal cases require in situ hybridization (FISH) to elucidate ERBB2 overexpression. If FISH testing confirmed an ERBB2/CEP17 (centromere enumeration probe for chromosome 17) ratio of 2 or higher in 50% or more cells, this is considered a positive result. 0+ and 1+ staining intensity are considered negative [36]. Authors from Japan, Korea, and the USA recently published a harmonization broadening provisional diagnostic criteria for ERBB2-positive mCRC. These criteria differ from those previously described in that the membrane staining positivity of a lower percentage of cells (10%) is taken into account. For example, a complete, lateral, or circumferential membrane staining with strong intensity and within >10% of tumor cells would be considered as IHC 3+, while an incomplete, lateral, or circumferential membrane staining with weak/moderate intensity and within >10% of tumor cells; or complete, lateral, or circumferential membrane staining with strong intensity and within ≤ 10% of tumor cells would be considered as IHC 2+. ERBB2 positivity was defined as IHC 3+ or IHC 2+/FISH positive [40]. ERBB2-positive, low expression has recently gained interest due to recent encouraging published results in breast cancer. Some authors suggest that IHC 2+/FISH-negative cases should be considered as ERBB2-low in CRC [41]. Implications for practice will be discussed afterward.

More recently, next-generation sequencing (NGS) has gained interest as an alternative technique to assess ERBB2 positivity as it can also provide information on ERBB2 and other oncogenic drivers’ mutational status. In a recent study, applying HERACLES criteria, IHC and NGS showed 92% concordance at the positive ERBB2 cutpoint and 99% concordance if equivocal cases were also considered positive. On the other hand, if ERBB2 IHC is treated as a screening tool, HERACLES-defined positive HER2 staining is 47% sensitive and 100% specific, whereas HERACLES-defined equivocal staining or greater is 93% sensitive and 100% specific for amplification by NGS [42]. Trying to harmonize IHC/FISH criteria, some authors suggest that CRC can be diagnosed as ERBB2+ with NGS if a copy number variant (CNV) of ≥ 5.0 is found in NGS, while CNV of 4.0 and 4.9 should be confirmed by IHC/FISH. This suggestion was validated in a retrospective cohort [40]. However, in a translational exploratory analysis in the HERACLES trial, the authors found that an ERBB2 copy number superior to 9.45 was predictive of response and progression-free survival [43]. Thus, more research is needed to find an optimal cut-off value for both diagnosis and prediction of benefit.

ctDNA (circulating tumor DNA) is becoming an attractive detection technique as it is less invasive than a conventional biopsy. This would allow repeating determinations during disease to track response and progression and to early detect the emergence of cellular clones resistant to therapy. This idea is also supported by evidence from a longitudinal tracking of ctDNA in the blood of patients included in the HERACLES trial. In this study, the dynamics of the presence of ERBB2 alleles increased in patients that were not responding to treatment and decreased in patients who had tumors that were responsive to treatment. Moreover, emerging KRAS mutant clones, BRAF amplification, mutations in ERBB2, and alterations in PI3KCA and PTEN appeared after progression to treatment with anti-ERBB2 agents. Some of these mutations had been previously linked to anti-ERBB2 resistance [44]. However, an adequate concordance between techniques is of capital importance before implementing the use of ctDNA to detect ERBB2 alterations. In an analysis of the HERACLES trial, ctDNA sequencing by Guardant360 assay correctly identified 96.6% of samples as ERBB2 amplified. Moreover, this study suggests a plasma copy number (pCN) ≥2.4 copies as a possible threshold representative of those patients whose HER2 amplification is the primary driver of malignancy. However, to improve the diagnostic performance of ctDNA, the authors developed an adjusted plasma copy number (apCN) in order to correct for variation in plasma tumor fraction between samples that can affect the tumor contribution to the circulating DNA pool. This apCN showed a stronger correlation than pCN (r = 0.86 vs. r = 0.52) between tissue HER2 copy number [45]. This approach was used in a later substudy of the TRIUMPH trial (discussed later) where apCN’s association with clinical benefit was similar between tissue and ctDNA NGS [46]. These observations suggest that serial determinations of ctDNA of patients treated with anti-ERBB2 therapies could be useful to monitor response to treatment and to elucidate resistance mechanisms and alternative therapeutic approaches.

However, there are points of debate regarding the diagnosis of ERBB2 positivity in CRC. The first one is the concordance between ERBB2 positivity between primary and metastatic lesions. Discordance rates seem to be relatively high, as a recent study using the HERACLES system suggests. In this study, the primary positivity rate was 11.2%, while in corresponding lymph nodes was 10.1% and 31.8% in liver metastases, showing a low concordance. However, this study has its limitations as no FISH was performed to confirm equivocal samples, and no information was given about the treatments the patient had received and their temporal relationship with sample collection. This could be relevant as ERBB2 could represent an acquired resistance mechanism of anti-EGFR treatments, as will be discussed afterward. There is also evidence that ERBB2 expression could be dynamic, with changes not only after treatment with anti-EGFR drugs but also with changes after anti-ERBB2 exposure as a resistance mechanism. In a patient included in the HERACLES trial, a warm autopsy protocol was applied, which allowed for the analysis of progressing hepatic lesions after treatment with trastuzumab and lapatinib. Two of the three progressive lesions were ERBB2 negative after the treatment, providing a biological rationale for the progression [44]. It is also not clear as to how chemotherapy (QT) treatment alone could influence ERBB2 expression. There is little retrospective evidence in this regard, with a study showing only 2.2% of 139 patients after chemoradiotherapy having ERBB2 overexpression in surgical specimens, lower than usually reported in the literature [34]. However, no information about ERBB2 status in previous biopsy specimens has been reported in this study, so we cannot conclude this low prevalence was only due to treatment.

## 4. Clinical Features of Patients with HER2-Positive mCRC

Evidence suggests that ERBB2 tumors are more common in the left side of the colon (including the rectum), although they may not be confined to the left side. This may be related to differences in organogenesis during embryonic development [47]. There is also evidence that canonical molecular subtype (CMS2) is enriched in ERBB2-positive tumors. CMS2 represents 37% of cases, with a greater prevalence of left-sided tumors with epithelial differentiation, alterations in WNT and MYC signaling, and more frequent copy number gains in oncogenes (including ERBB2) [48,49]. Preclinical data may suggest that CMS2 tumors are more responsive to EGFR and ERBB2 blockade by tyrosine kinase inhibitors than the other subtypes [50].

There is also evidence of a different pattern of dissemination in patients with ERBB2-positive disease. In a retrospective cohort of CRC patients with resected brain metastases, up to 12% had IHC 3+ for ERBB2, which is higher than expected according to the reported prevalence of ERBB2-positive primaries [51]. The development of central nervous system (CNS) metastases has also been linked with treatment with trastuzumab and lapatinib in the HERACLES trial. CNS progression appeared in up to 19% of patients treated in this trial, a high prevalence compared to historical series [52]. There is also evidence linking ERBB2 positivity with a higher probability of developing lung metastases [22] and ovarian metastases [53]. ERBB2 positivity in ovarian metastases was also correlated with the presence of peritoneal metastases [53].

Regarding ERBB2 as a prognostic factor, evidence is conflicting. Older studies found associations of ERBB2 positivity with worse overall survival (OS) and worse stage at diagnosis; however, these studies considered cytoplasmic staining as well as membranous staining. These methods contrast with modern diagnostic criteria, so these results are difficult to interpret [38,54]. In this regard, in a modern and large (1654 patients) primary colorectal cancer study, ERBB2 positivity (1.6%; 26 patients) was associated with advanced stages and a non-significant tendency towards worse OS [55]. Furthermore, a post-hoc analysis of the PETACC-8 trial (1795 patients) found that stage-III ERBB2 positive and ERBB2 exon 19–21 mutated patients (2.9%; 49 patients, and 1%; 17 patients, respectively) had a shorter time to recurrence and worse OS, and that observation was maintained after adjusting for other adverse prognostic factors as KRAS mutation [56]. In another study, ERBB2-low patients were found to be more frequent than ERBB2 positive, with a significantly better prognosis in terms of OS than ERBB2-positive patients (33.3 months vs. 18.2 months), and in terms of PFS (2.2 months vs. 7.8 months) [41]. Another study found that nuclear staining of cyclooxygenase-2 (COX-2) correlated with high ERBB2 staining in colorectal patients. Negative nuclear staining of COX-2 and low ERBB2 staining correlated with a better prognosis than high nuclear staining of COX-2 and high ERBB2 membrane staining. However, this study did not use the above-mentioned diagnostic criteria for ERBB2 positivity, and hence its results are difficult to extrapolate [57].

On the other hand, a large cohort of 3256 patients were included in the QUASAR (adjuvant trial, stage I, II, and III patients) PICCOLO and FOCUS (metastatic patients) trial. In this cohort, 2.2% (*n* = 29) of stage IV patients and 1.3% (*n* = 25) of stage II and III patients were found to have ERBB2 positivity by IHC. There was no significant correlation between ERBB2 and recurrence or overall survival [35]. Furthermore, a German study that included 264 patients found that ERBB2 positivity (26–7%; 60 patients) was associated with better disease-free survival (DFS). This study used diagnostic criteria with a low cut-off value, which explains the high proportion of ERBB2-positive patients [58]. As different methodologies were used in the aforementioned studies, as well as in the inconsistent results found, there is no current consensus on the role of ERBB2 as a prognostic factor in CRC. Table 1 summarizes the main studies about the prognostic significance of ERBB2 in CRC.

ERBB2 has also been proposed as a marker of resistance to anti-EGFR therapies. A preclinical study has suggested that ERBB2 amplification could mediate anti-EGFR primary resistance in xenograft models, particularly in KRAS/NRAS/BRAF/PI3KCA wild-type patients. Importantly, this resistance to cetuximab could be reversed with a combined inhibition of ERBB2 and EGFR [60]. Another study found evidence of ERBB2 amplification in ctDNA in patients primarily resistant to anti-EGFR therapy [61]. A retrospective study suggested that ERBB2 patients were less likely to respond to anti-EGFR therapies. However, this reduction in response rates was not directly correlated with survival. This study found a non-significant trend to worse progression-free survival and no significant differences in OS [22]. A study focused on ERBB2-low patients found a significant difference in progression-free survival (PFS) between ERBB2-low and ERBB2-positive patients treated with anti-EGFR agents (7.8 m vs 2.2 m) [41]. Other experiments and studies have also suggested that HER2 could represent a mechanism of acquired resistance to antiEGFR therapies. The introduction of ERBB2 in cells that were previously sensitive to cetuximab conferred resistance to this drug by causing abnormal activation of ERK1/2 [62]. Another study of ctDNA in patients previously treated with anti-EGFR therapies showed amplification of the ERBB2 gene upon progression in 22% of patients. In this study, a patient who progressed on cetuximab had a progressive lesion rebiopsied, showing evidence of HER2 overexpression. Notably, ERBB2 amplification was absent in primary tumor [63]. However, to date, there are no specific recommendations regarding the role of ERBB2 to guide therapeutic decisions on anti-EGFR therapies or the role of rebiopsy in progression.

## 5. Clinical Trials for Patients with ERBB2-Positive mCRC

There are several types of therapies that target ERBB2: monoclonal antibodies, tyrosine kinase inhibitors (TKIs), and antibody–drug conjugates (ADCs). Table 2 summarizes clinical trials for patients with ERBB2-positive mCRC and the presented results.

### 5.1. Monoclonal Antibodies

Trastuzumab and pertuzumab are monoclonal antibodies that target ERBB2. They bind to the extracellular domains of the receptor, inhibiting dimerization and promoting antibody-dependent cellular cytotoxic effects [74,75].

Initial trials investigated the combination of trastuzumab with QT. A phase II clinical trial evaluated trastuzumab and FOLFOX as the second or third line of treatment in ERBB2-positive patients. The ORR (overall response rate) was 24%, and the median duration of response was 4.5 m (2.7–11) [64]. Ramanathan et al. led a phase II clinical trial that evaluated trastuzumab plus irinotecan in ERBB2-positive mCRC patients. ERBB2 overexpression was detected in 8% of screened patients by IHC. Nine patients were included, and partial responses were seen in five of seven evaluable patients. However, they concluded that the low rate of ERBB2 overexpression limited more investigations in mCRC [65].

Several clinical trials have evaluated the effectiveness of the combination of two ERBB2-directed monoclonal antibodies. MyPathway was a phase IIa, multiple-basket, clinical trial that evaluated the combination of trastuzumab and pertuzumab in 57 patients with pretreated ERBB2-positive mCRC. mPFS was 2.9 m, mOS was 11.5 m, and ORR was 32%. The most common treatment-emergent adverse events were diarrhea (33%), fatigue (32%), and nausea (30%). ORR was 40% in patients with KRAS WT (wild-type) and 8% in patients with KRAS mutated tumors, so KRAS status was associated with anti-ERBB2 therapeutic efficacy [66].

TAPUR was a phase II basket clinical trial that investigated the addition of trastuzumab to pertuzumab in 28 pretreated patients with mCRC and ERBB2 overexpression/amplification. ORR was 14%. Differences in ORR compared to other studies might be explained by the inclusion of patients with concomitant RAS variations (additional analyses by RAS mutation status are pending). Two patients had at least one grade III adverse event related to trastuzumab and pertuzumab, including anemia, infusional reactions, and left ventricular dysfunction [67].

TRIUMPH was a phase II clinical trial that enrolled patients with RAS WT mCRC and ERBB2 amplification detected in tissue or ctDNA. mPFSs were 4.0 m and 3.1 m in patients with ERBB2-positive tissue and ctDNA (Guardant360), respectively. ORRs were 30% and 28% in patients with ERBB2-positive tissue and ctDNA, respectively. Patients without ctDNA variations in RAS/BRAF/PIK3CA/ERBB2 had a better response than those with a ctDNA variation in one of these genes: ORR was 44% vs. 0% in ERBB2-positive tissue and 37% vs. 0% in ERBB2-positive tissue ctDNA, respectively. mPFSs were 4.0 m (1.4–5.6) and 3.1 m (1.4.5.6) in patients with ERBB2-positive tissue and ctDNA, respectively, whereas mOSs were 10.1 (4.5–16.5) and 8.8 m (4.3–12.9) in patients with ERBB2-positive tissue and ctDNA, respectively. TRIUMPH demonstrated that decreased ctDNA fraction and ERBB2 plasma copy number three weeks after treatment initiation correlated with treatment response [46,68].

### 5.2. Tyrosine Kinase Inhibitors (TKIs)

Lapatinib, pyrotinib, tucatinib, and neratinib are oral TKIs. They inhibit the intracellular tyrosine kinase domain and phosphorylation of the ERBB2 receptor, inhibiting cell growth [76]. Several clinical trials have evaluated the efficacy of dual ERBB2 inhibition through the combination of trastuzumab and TKI.

HERACLES-A was a non-randomized, open-label, phase II clinical trial where treatment-refractory KRAS WT ERBB2-positive mCRC patients were treated with trastuzumab and lapatinib. A total of 914 patients were screened, and 48 patients (5%) were identified as ERBB2-positive (IHC 3+ in ≥ 50% of cells or IHC 2+ and an ERBB2:CEP17 ratio > 2 in more than 50% of cells by FISH). However, only 27 patients were eligible for the trial. In total, 74% of patients previously received at least four lines of treatment. None of the 15 patients who were evaluable for prior response to anti-EGFR therapy had obtained an objective response with cetuximab or panitumumab. Six patients (22%) had grade III adverse events: fatigue in four patients, skin rash in one patient, and increased bilirubin concentration in one patient [43].

Long-term follow-up analysis of the HERACLES-A study shows that 35 patients received trastuzumab and lapatinib, ORR was 28%, mPFS was 4.7 m (95% CI: 3.7–1), and mOS was 10.0 m (95% CI: 7.9–15.8). Progression in CNS occurred in 19% of patients [69].

Yuan et al. led a phase II clinical trial of 11 patients with ERBB2-positive mCRC treated with trastuzumab and pyrotinib. The ORR was 27%, 50% in patients with KRAS wild-type mCRC, and 60% in patients with RAS wild-type disease. Diarrhea was the most common grade III adverse event (73%), causing dose interruption and reduction in 64% and 45% of patients, respectively [70].

The MOUNTAINEER study was an open-label, single-arm phase II clinical trial when 23 pre-treated RAS WT ERBB2-positive mCRC received trastuzumab and tucatinib. The ORR was 55%, mPFS was 6.2 m (95% CI: 3.5–NE), and mOS 17.3 m (95% CI: 12.3–NE). The grade III adverse events were low (8%) [71].

Jacobs et al. reported the results of a phase Ib clinical trial involving 11 patients with RAS/BRAF/PIK3CA WT; ERRB2-positive tumors were treated with neratinib and cetuximab. However, it did not show responses: seven received stable disease, four of whom had ERBB2 amplification either in the primary tumor or the enrolment biopsy [77].

### 5.3. Antibody–Drug Conjugates (ADCs)

Trastuzumab emtansine (T-DM1) and trastuzumab deruxtecan (TD) are ADCs. Whereas trastuzumab is linked to a microtubule inhibitor in T-DM1, trastuzumab is joined to topoisomerase inhibitor. If the trastuzumab binds ERBB2, the ADC is internalized, the linker is cleaved, and a cytotoxic effect is made [78].

The HERACLES-B trial was a single-arm, phase II clinical trial that investigated the combination of pertuzumab and T-DM1 in RAS/RAF WT ERBB2-positive mCRC patients refractory to standard treatments. A total of 31 patients were enrolled. The primary endpoint of the study was ORR, being negative for this endpoint (9.7%, 95% CI: 0–28). However, 21 patients (67.7%) had stable disease resulting in a disease control rate of 77.4%. mPFS was 4.1 m (95% CI: 3.6–5.9); this result was similar to the HERACLES-A study. Grade III adverse events were observed in two patients (thrombocytopenia), and the most frequent grade II adverse events were nausea and fatigue [72].

The DESTINY-CRC01 trial was a phase II clinical trial that evaluated TD in treatment-refractory patients with RAS/BRAF V600E WT ERBB2-positive mCRC. Patients were enrolled in one of three cohorts on the basis of the level of ERBB2 amplification to explore the association of ERBB2 expression with the activity of TD in mCRC: cohort A (ICH3-positive or ICH2-positive and FISH-positive), cohort B (IHC2-positive and ISH-negative), and cohort C (IHC1-positive). All patients received TD 6.3 mg/kg every three weeks [73]. This dose was the same as recommended for gastric cancer and higher than breast cancer, and it was chosen because of previous studies of pharmacokinetics and antitumor activity [79,80,81,82].

A total of 78 patients were enrolled in the DESTINY-CRC01: 53 in cohort A, 7 in cohort B, and 18 in cohort C. The ORR was 45.3% in cohort A: 57.5% in patients that were ERBB2 ICH3 positive and 7.7 in patients that were ERBB2 ICH2 positive/ISH-positive. ORR was 0% in cohorts B and C. mPFS was 6.9 m, 2.1 m, and 1.4 in cohort A, cohort B, and cohort C, respectively. mOS was 15.5 m, 7.3 m, and 7.7 m in cohort A, cohort B, and cohort C, respectively. However, grade III or worse adverse events that occurred in at least 10% of all patients were decreased neutrophil count and anemia. Five patients had interstitial lung disease or pneumonitis (two grade 2; one grade 3; two grade 5, the only treatment-related deaths). A higher clinical response was detected with higher plasma ctDNA ERBB2 copy number. Antitumor activity was observed in patients regardless of ctDNA-detected activating RAS or PIK3CA mutations [83].

### 5.4. Ongoing Clinical Trials and Novel Anti-ERBB2 Therapies

Several ongoing clinical trials are exploring anti-ERBB2 therapies that evaluate the efficacy of small molecule inhibitors, ADCs, and their combination with established therapies [84].

The MATCH trial is a clinical trial of targeted therapy diagnosed by genetic testing in solid tumors or lymphomas after progression of at least one line of treatment. Two cohorts of patients with ERBB2-amplified tumors are treated with trastuzumab plus pertuzumab (cohort J) or T-DM1 (cohort Q) [85].

The MOUNTAINEER trial has been expanded to include a cohort of tucatinib monotherapy (NCT03943313) [86]. NSABP FC-11 is a three-cohort, phase II clinical trial in patients with RAS/BRAF/PIK3CA WT ERBB2-positive mCRC. This study compares the efficacy of neratinib and trastuzumab (Arm-1: patients who have ERBB2 amplification and prior anti-EGFR treatment or ERBB2 mutation with or without prior anti-EGFR treatment) vs. neratinib plus cetuximab (Arm-2: patients who are ERBB2 non-amplificated or ERBB2 amplification without prior anti-EGFR treatment) (NCT03457896) [87]. The first results of NSABP FC-11 were presented at the ASCO 2022 meeting [88]. Arm-1 closed due to poor accrual, and those patients have been excluded from further analysis. Arm 2 enrolled 21 patients with 15 evaluable for response by imaging. Of the 15 evaluable patients, there were 6 PR, 5 of 13 ERBB2 non-amplification, 1 of 2 ERBB2 amplification, and 5 SD. The ORR in all patients who were treated with at least one dose was 33%. Common grade 3–4 were diarrhea (24%), rash (8%), and abdominal pain/distension (8%).

Following the results of DESTINY-CRC01, DESTINY-CRC02 is a phase II clinical trial that is going to determine the efficacy and safety of TD in patients ERBB2-positive at 5.4 mg/kg and 6.4 mg/kg doses [89]. The dose of 5.4 mg/kg has not been tested in ERBB2-positive mCRC patients, but this dose has shown efficacy in other tumors [79,80,81].

Several trials explore the role of anti-ERBB2 therapies in earlier lines of treatment compared to QT. The MODUL trial is a randomized, open-label, parallel-group study that evaluates the efficacy and safety of biomarker-driven maintenance treatment in the first line of treatment in mCRC, including an ERBB2-positive cohort (capecitabine, trastuzumab, and pertuzumab) (NCT02291289). SWOG study (S1613) is a multicenter, randomized, phase II clinical trial that tries to compare the efficacy of trastuzumab plus pertuzumab vs. cetuximab plus irinotecan in patients with RAS/RAF WT ERBB2-positive mCRC (NCT03365882). Patients have to have been treated with at least one prior line of therapy for mCRC that did not include anti-EGFR or anti-ERBB2 agents.Zanidatamab (ZW25) is a bispecific antibody that binds to two different regions on the ERBB2 receptor, increasing antibody binding density and improving receptor internalization and downregulation. It is used in phase I and II clinical trials in patients with ERBB2-positive gastrointestinal cancers, including mCRC (NCT02892123, NCT03929666) [90]. A166 uses an antibody with the same amino acid sequence as trastuzumab and it is linked to duostatin-5. Safety profile of A166 has been observed in a phase I clinical trial. ZW49 has an auristatin payload conjugated to the antibody ZW25, which binds to the same ERBB2 domains as trastuzumab and pertuzumab. ZW49 is being evaluated in a phase I clinical trial (NCT03602079).

A phase I clinical trial will investigate the efficacy and safety of two chimeric (trastuzumab-like and pertuzumab-like) ERBB2 vaccines in patients with various metastatic solid tumors, including mCRC (NCT01376505). Another phase I trial uses an allogenic-donor-derived natural killer (NK) cell cancer immunotherapy (FATE-NK100) as monotherapy or in combination with trastuzumab or cetuximab in multiple ERBB2-positive tumors (NCT03319459). An anti-ERBB2 chimeric antigen receptor (CAR)-modified T cell therapy is evaluated in several ERBB2-positive solid tumors, including mCRC (NCT02713984) [91]. Moreover, HER2-AdVST (ERBB2 chimeric antigen receptor-modified adenovirus-specific cytotoxic T lymphocytes) joined to an intra-tumor injection of CAdVEC (an oncolytic adenovirus that helps the immune system) is being evaluated in an ongoing clinical trial (NCT03740256). Other clinical trials are trying to show the effectiveness of peptide vaccines (NCT01376505). Patients receive an ERBB2/neu peptide vaccine comprising measles virus epitope MVF-ERBB2-2 (266–296) and MVF-ERBB2 (597–626) emulsified with nor-MDP in ISA 720 intramuscularly.

## 6. Discussion

The results of clinical trials targeting ERBB2 positivity in mCRC have shown promising results in ORR and PFS, especially when standard treatments have been administrated. This demonstrates the importance of the diagnosis of target molecular biomarkers in the era of precision medicine. Whereas the ORR of these clinical trials are 10–40%, the trifluridine/tipiracil (TAS-102) and regorafenib have ORR of 2% and 1%, respectively [7,8]. Table 3 shows the salient points of this review.

Although some results of the clinical trials shown above are not definitive, they illustrate the necessity of the development of phase III clinical trials in ERBB2-positive mCRC patients. These phase III clinical trials have to try to answer some important clinical questions: Which is the better sequence of treatment, starting with targeted therapy or standard treatment? Which targeted therapy is better? Is sequential targeted therapy relevant in ERBB2-positive mCRC?

We do not have results that suggest which is the better sequence of treatment in ERBB2-positive mCRC because the majority of the described clinical trials are realized in patients in whom standard treatment fails [92,93]. We do not know which targeted therapy shows better ORR or mPFS. The management of adverse events of targeted therapy is well known—for example, the appearance of left ventricular dysfunction with trastuzumab or interstitial lung disease when TD is used. However, some results suggest that sequential therapy may be relevant in ERRB2-positive mCRC because 30% of patients in DESTINY-CRC01 were previously treated with other anti-ERRB2 therapies.

In an ideal scenario, when a patient is diagnosed with mCRC, a biopsy has been analyzed by a pathologist and IHC and NGS should be realized. Moreover, a liquid biopsy of ctDNA should be realized. These procedures should be repeated when a line of treatment fails. This strategy would give a lot of molecular information, such as the development of mechanisms of resistance. In this way, a medical oncologist would be able to select the best treatment, including the inclusion in a clinical trial. However, this approach is not useful. Firstly, performing multiple biopsies carries certain risks. Secondly, NGS and ctDNA in mCRC are under research at this moment. The information obtained from NGS and ctDNA is sometimes difficult to integrate and understand. Third, making these procedures is expensive, and in a public health system, the government could deny the payment because this strategy may not be efficient enough (we are looking for target rare molecular biomarkers to start expensive targeted therapies). This could also apply for private insurances.

There is controversy about when the medical oncologist must look for rare target molecular biomarkers. Some oncologists think that rare molecular biomarkers should be determined before starting the first line of treatment. Patients have a good clinical status at this moment, and targeted therapies have better ORR and PFS than QT, as in other pathologies (EGFR mutations in non-small cell lung cancer). On the other hand, other oncologists affirm that the determination of rare molecular biomarkers must be performed when patients maintain good clinical status and standard treatments have failed.

The guidelines of treatment in mCRC show that the determination of RAS (KRAS/NRAS) mutations, BRAF mutations, and deficient mismatch repair should be realized before starting the first line of treatment (6,32). The ESMO (European Society of. Medical Oncology) guidelines do not mention ERBB2 amplification/overexpression [6]. The NCCN (National Comprehensive Cancer Network) guidelines state that testing ERBB2 amplification/overexpression should be made in patients with mCRC and absence of RAS or BRAF mutation. ERBB2-targeted therapies are recommended as subsequent therapy options, encouraging enrollment in a clinical trial (32). The actualization of these guidelines should define the optimal timing and technique for testing, the most adequate panel, and whether all RAS WT mCRC should be tested for ERBB2 [94].

The search for target rare molecular biomarkers illustrates the complexity of precision medicine, so it is required that a medical oncologist has to study molecular biology and clinical treatments. Moreover, it shows the necessity of multidisciplinary work. In our center, we have to work with other specialists to obtain tissue for the molecular diagnosis, and we have a fluid relationship with pathologists. If we obtain a target rare molecular biomarker, we must discuss it with other medical oncologists when we want to include a patient in a clinical trial.

## 7. Conclusions

The management of mCRC in the era of precision medicine is becoming more complex. Amplification of ERBB2 is present in 3% of patients with mCRC and 5% of patients with RAS and BRAF wild type. Several clinical trials have demonstrated that the ERBB2 receptor represents a good option for targeted therapy in mCRC and may represent an option when standard treatments fail to control mCRC. However, therapies are currently not approved for these patients, and the recommendation is the enrollment of patients in a clinical trial.

## Figures and Tables

**Figure 1 cancers-14-03718-f001:**
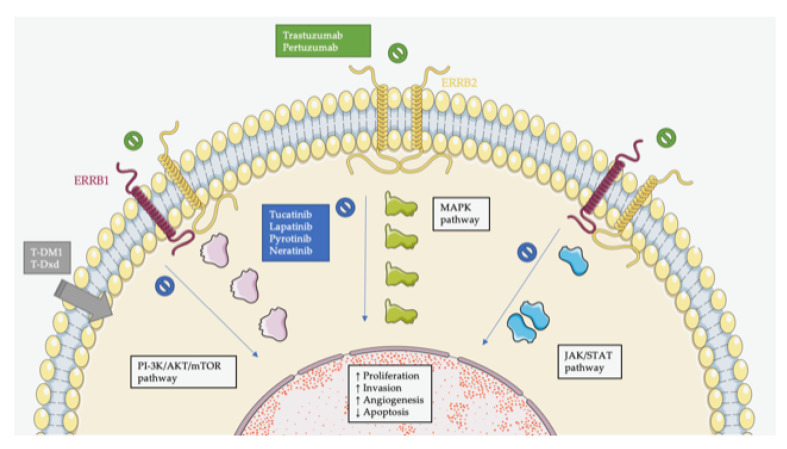
Molecular biology of HER2 receptor and mechanisms of action of main available drugs. Activation of HER2 by overexpression (enabling uncontrolled homo- or heterodimerization) or by activating mutations leads to constitutive activation of MAPK, PI-3K/AKT/mTOR, Src, and JAK/STAT pathways. Available drugs block this activation by inhibition of the dimerization or by inhibition of the tyrosine kinase domain of the receptor. T-DM1 and T-Dxd exert their cytopathic effects by liberation of chemotherapy in high concentrations in tumors expressing HER2.

**Table 1 cancers-14-03718-t001:** Main studies on the prognostic significance of ERBB2 positivity.

**Study (Year)**	**Patients**	**Stage**	**ERBB2 Positivity** **Criteria**	**Prognostic** **Significance**
Yagisawa et al. (2021) [41]	370	IV	International harmonization	Better prognosis of ERBB-low patients
Sawada et al. (2018) [33]	359	I–IV	HERACLES	No differences in OS
Park et al. (2018) [34]	145	I–III	Modified HERACLES	No differences in survival
Richman et al. (2016) [35]	3256	I–IV	Gastric cancer scoring	No differences in OS or PFS
Laurent-Puig et al. (2016) [56]	1804	III	HERACLES + NGS	Lower DFS and OS
Heppner et al. (2014) [55]	1645	I–IV	Gastric cancer scoring	No significant trend to poorer OS
Conradi et al. (2013) [58]	264	II–IV	Gastric cancer scoring	Better DFS
Kruszewsky et al. (2010) [59]	202	I–IV	Membranous + cytoplasmic staining	No association with OS
Osako et al. (1998) [38]	146	Dukes A-D	Membranous + cytoplasmic staining	Poorer survival in cytoplasmic staining
Kapitanovic et al. (1997) [54]	221	Bening, premalignant and malignant lesions	Membranous staining	Strong staining correlates with poorer survival

Abbreviations: DFS: disease-free survival; NGS: next-generation sequencing; OS: overall survival.

**Table 2 cancers-14-03718-t002:** Clinical trials targeting ERBB2-positive mCRC.

Trial	Reference	Treatment	n	Prior Lines of Treatment	Mutational Status	mPFS (m)	ORR (%)
Trastuzumab + QT							
Clark et al.	[64]	Trastuzumab + FOLFOX		<2	NS	NR	24
Ramanathan et al.	[65]	Trastuzumab + irinotecan	9	≤1	NS	NR	71
Monoclonal antibodies							
MyPathway	[66]	Trastuzumab + pertuzumab	57	≥1	RAS WT	2.9	32
TAPUR	[67]	Trastuzumab + pertuzumab	28	≥0	NS	NR	14
TRIUMPH	[68]	Trastuzumab + pertuzumab	27 (Tissue)	≥1	RAS WT	4.0	30
			25 (ctDNA)			3.1	25
Monoclonal antibody + TKI							
HERACLES-A	[43,69]	Trastuzumab + lapatinib	35	≥2	KRAS WT	4.7	28
Yuan et al.	[70]	Trastuzumab + pyrotinib	11	≥2	RAS WT and mutated	NR	27
MOUNTAINEER	[71]	Trastuzumab + tucatinib	23	≥2	RAS WT	8.1	52
ADCs							
HERACLES-B	[72]	Pertuzumab + T-DM1	31	≥2	RAS/BRAF WT	4.1	10
DESTINY-CRC01	[73]	TD	53 (Cohort A)	≥2	RAS/BRAF WT	6.9	45

Abbreviations: ADCs: antibody–drug conjugates, ctDNA: circulating tumor DNA, m: month, mPFS: median progression-free survival, NR: not reported, NS: not specified, ORR: overall response rates, QT: chemotherapy, T-DM1: trastuzumab emtansine, TD: trastuzumab deruxtecan, TKI: tyrosine kinase inhibitor, WT: wild type.

**Table 3 cancers-14-03718-t003:** Salient points of the review.

Molecular biology
The best-known pathogenic mechanisms involved in ERBB2 aberrant activation are overexpression of ERBB2 and activating mutations.
Diagnosis of HER2-positivity in mCRC
The HERACLES diagnostic criteria are nowadays the diagnostic criteria most commonly used, although not the only ones described in the literature: ○ Positive: intense (3+) expression in ≥50% of cells. ○ Equivocal: moderate (2+) expression in ≥50% or 3+ ERBB2 in more than 10% but less than 50% of tumor cells. FISH must be performed, with an ERBB2/CEP17 ratio of 2 or higher in 50% or more cells, considered a positive result. ○ Negative: 0+ and 1+ staining. NGS could represent an alternative diagnostic technique, but adequate threshold positivity must be defined. ctDNA is a promising less-invasive diagnostic technique but needs to be validated.
Clinical features of patients with HER2-positive mCRC
ERBB2-positive tumors are more common in the left side of the colon. CMS2 is enriched in ERBB2-positive tumors. Regarding ERBB2 as a prognostic factor, evidence is conflicting. ERBB2 has also been proposed as a marker of resistance to anti-EGFR therapies, innate or acquired.
Clinical trials for patients with ERBB2-positive mCRC
MyPathway, TAPUR, and TRIUMP were phase II clinical trials that have evaluated the effectiveness of the combination of two ERBB2-directed monoclonal antibodies (trastuzumab and pertuzumab). Several clinical trials have evaluated the paper of dual ERBB2 inhibition by the combination of trastuzumab and TKI: HERACLES-A (lapatinib) and MOUNTAINEER (tucatinib), showing promising ORR. The HERACLES-B clinical trial used the combination of pertuzumab and T-DM1, and the DESTINY-CRC01 clinical trial used trastuzumab-deruxtecan. They showed an important ORR. Several ongoing clinical trials are exploring the efficacy of small molecule inhibitors, ADCs, and their combination with established therapies or the role of anti-ERBB2 therapies in earlier lines of treatment compared to QT. However, therapies are currently not approved for these patients, so the enrollment of patients in a clinical trial is recommended.

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
