# Peer review of "Precision Medicine in Metastatic Colorectal Cancer: Targeting ERBB2 (HER-2) Oncogene"

_cancers, 2022, doi:10.3390/cancers14153718_

Round 1

Reviewer 1 Report

The manuscript titled 'Precision Medicine in metastatic colorectal cancer: targeting 2 ERBB2 (HER-2) oncogene' provides an in depth account of CRC. Please see some suggestions below:

1. The abbreviations need to expanded first time these are mentioned in the review. 

2. Change sentence - 25 % of CRC patients have 39 metastatic lesions at diagnosis and almost 50 % of patients with early-stage CRC will dis- 40 seminated disease

3. line 47 'such as'

4. Some unnecessary capital letters. Need to be consistent with the words 'precision medicine'.

5. Please make a figure to describe the molecular biology as it would help a wide audience of readers including early career to understand the perturbations as a consequence of constitutive activation of various pathways.

6. Need to refer to the section in line 141,176 and 190

7. Please read the manuscript to check for sentences that have word(s) missing e.g. 510-512 and 512-514

8. Please highlight the salient points of the review in a table or flowchart.

9. Just a suggestion if pictures can be added from published articles to make some of the content more meaningful and to contextualise the results described about IHC staining etc.  

Author Response

Madrid, 25th July 2022

Dear reviewer 1

We are pleased to submit for publication the revised version of the manuscript. At the following, the pints mentioned will be discussed:

1. The abbreviations need to expanded first time these are mentioned in the review. 

Thanks for this remark. We have reviewed abbreviations and they have been expanded first time these are mentioned in the review.

2. Change sentence - 25 % of CRC patients have 39 metastatic lesions at diagnosis and almost 50 % of patients with early-stage CRC will dis- 40 seminated disease.

Thanks for this remark. We have changed it.

3. line 47 'such as'.

Thanks for this remark. We have changed it.

4. Some unnecessary capital letters. Need to be consistent with the words 'precision medicine'.

Thanks for this relevant comment. We have changed them.

5. Please make a figure to describe the molecular biology as it would help a wide audience of readers including early career to understand the perturbations as an consequence of constitutive activation of various pathways.

We have added a figure to describe the molecular biology.

6. Need to refer to the section in line 141,176 and 190.

We don’t understand this sentence. The reference of line 141 is 41 (Yagisawa et al, 2021), the reference of line 176 is 45 (Siravegna et al. 2018) and the reference of line 190. The reference of the line 190 also appears.

7. Please read the manuscript to check for sentences that have word(s) missing e.g. 510-512 and 512-514

Thanks for this remark. We have read it and we have corrected it.

8. Please highlight the salient points of the review in a table or flowchart.

We have added the salient points of the review in a table.

9. Just a suggestion if pictures can be added from published articles to make some of the content more meaningful and to contextualise the results described about IHC staining etc.  

We are going to try this question. The principal problem is the copyright of the images.

Thank you for the revision. Kind regards.

Javier Torres

Reviewer 2 Report

For the authors

Overexpression of the ERBB2 tyrosine kinase receptor and a member of the epidermal growth factor receptor family, results in aberrant cell migration, growth, adhesion, and differentiation. ERBB2-targeted therapeutic options are available for patients with ERBB2-positive breast and gastric/gastroesophageal tumors. Although 3% of patients with metastatic colorectal cancer, have ERBB2-positive tumors, no approved ERBB2-targeted therapies are available to them.

Torres-Jiménez et al reviewed recent clinical trial data investigating the efficacy of treatments targeting ERBB2 in CRC patients with ERBB2 positive tumors, including monoclonal antibodies, tyrosine kinase inhibitors, antibody-drug conjugates, and combinations with common chemotherapies. A number of these studies revealed favourable outcomes. Despite this, these treatments are not yet adopted into the clinic because, as the authors outline there are several issues that must be resolved including patient selection based on CRC subtype, previous therapies, presence or absence of other contributing mutations and appropriate treatment regimens. Thus, more research is needed before these treatments become part of the treatments available.

The authors covered the relevant studies adequately, however there is a recent review on this topic by Strickler et al 2022, which is not included. The authors need to include this in their revised manuscript and comment on how their publication adds to the literature.

Specific points

It would be helpful if they define the monoclonal antibody targets. For example, what are the targets of the monoclonals used treat deficient mismatch repair/microsatellite tumors?

The manuscript needs careful proofreading and thorough editing. There are too many expression errors and spelling/ grammatical errors that often introduce ambiguity. Below I highlight a few of these.

The first paragraph of section 4 needs to be reworked. First define the CMS2 subtype and then report on ERBB2 positivity and response to treatment.

Line 237 cyclooxygenase -2 is misspelled.

Line 325   check TRIUMP spelling

Line 345 it should be Yuan et al, led

Line 382 the % sign is missing

Line 436 should the word safety be used instead of security?

Last sentence line 447    specify what are the peptides used. Are they ERBB2 sequences and are they from mutated sections?

Discussion

Lines 451-542  Do you mean when given with standard treatments?

Line 464   where should used instead of were

Line 466 apparition is not the appropriate word

Author Response

Madrid, 25th July 2022

Dear reviewer 2

We are pleased to submit for publication the revised version of the manuscript. At the following, the pints mentioned will be discussed:

1. The authors covered the relevant studies adequately, however there is a recent review on this topic by Strickler et al 2022, which is not included. The authors need to include this in their revised manuscript and comment on how their publication adds to the literature.

The review written by Stricker et al (2022) has been included and it appears as reference number 29.

2. It would be helpful if they define the monoclonal antibody targets. For example, what are the targets of the monoclonals used treat deficient mismatch repair/microsatellite tumors?

We have defined the monoclonal antibody targets in introduction. The target of the monoclonal used treat deficient mismatch repair/microsatellite tumors are PD-1 (nivolumab) and CTLA4 (ipilimumab).

3. The manuscript needs careful proofreading and thorough editing. There are too many expression errors and spelling/grammatical errors that often introduce ambiguity. Below I highlight a few of these.

We have made an extensive editing on English language.

4. The first paragraph of section 4 needs to be reworked. First define the CMS2 subtype and then report on ERBB2 positivity and response to treatment.

Thanks for this remark. We have changed it.

5. Line 237 cyclooxygenase -2 is misspelled.

Thanks for this remark. We have changed it.

6. Line 325   check TRIUMP spelling.

Thanks for this remark. We have changed it.

7. Line 345 it should be Yuan et al, led.

Thanks for this remark. We have changed it.

8. Line 382 the % sign is missing

Thanks for this remark. We have changed it.

9. Line 436 should the word safety be used instead of security?

Thanks for this remark. We have changed it.

10. Last sentence line 447 specify what are the peptides used. Are they ERBB2 sequences and are they from mutated sections?

Patients receive a HER2/neu peptide vaccine comprising measles virus epitope MVF-HER-2 (266-296) and MVF-HER-2 (597-626) emulsified with nor-MDP in ISA 720 intramuscularly (IM) on day 1. More information about this clinical trial can be read in https://clinicaltrials.gov/ct2/show/NCT01376505.

11. Lines 451-542  Do you mean when given with standard treatments?

Standard treatments are described in introduction: several modern combinations of treatments (5-fluouracil (5-FU)/leucovorin (LV)/oxaliplatin (FOLFOX), 5-FU/LV/irinotecan (FOLFIRI), 5-FU/LV/oxaliplatin/irinotecan (FOLFOXIRI), targeted therapies as anti-epidermal growth factor receptor (EGFR), such as cetuximab and panitumumab, and anti-vascular endothelial growth factor (VEGF), such as bevacizumab and aflibercept, have been developed recently and they are used in first and second line of treatment of mCRC. Other therapies (trifluridine/tipiracil, regorafenib, raltitrexed) are used in third and successive lines of treatment.

13. Line 464 where should used instead of were

Thanks for this remark. We have changed it.

14. Line 466 apparition is not the appropriate word

Thanks for this remark. We have changed it.

Thank you for the revision. Kind regards.

Reviewer 3 Report

The review proposed by Dr. Ferreiro-Monteagudo and colleagues is focused on the use of precision medicine in metastatic colorectal cancer. In particular, the authors addressed several aspect concerning the ERBB2 oncogene: molecular structure of ERBB2 receptor, diagnosis of ERBB2 alteration (amplification or activating mutations), clinical features of the patients and clinical trials with ERBB2 targeted therapies.

The overall subject is one of considerable interest and I appreciated the effort of the authors in revised eligible and recent publications.

Moreover, this review evaluated an extensive number of published studies.

The manuscript is well written and comprehensive and English language and style are fine.

Minor suggestions:

-   In the “Simple Summary” session and in the “Abstract” session there are a lot of redundant information and often the same phrases are present in both the sessions. Please adjust this issue eliminating the redundant parts.

Author Response

Madrid, 25th July 2022

Dear reviewer 3

We are pleased to submit for publication the revised version of the manuscript. At the following, the pints mentioned will be discussed:

1. In the “Simple Summary” session and in the “Abstract” session there are a lot of redundant information and often the same phrases are present in both the sessions. Please adjust this issue eliminating the redundant parts.

Thanks for this remark. We have reviewed Simple Summary and Abstract and the they have been modified.

Thank you for the revision. Kind regards.

Javier Torres

Round 2

Reviewer 2 Report

The manuscript did benefit from the editing revision.

I picked up a couple of additional errors that need changing as follows

1 line 86  ER2 should be ERBB2

2 line 335 ....the paper of dual ERBB2.... the word paper should be efficacy I think 

3 'The MOUNTAINEER study was an open-label, single-arm phase was a phase 2 clinical trial when' ....line 355 needs to be rephrased.

4 line 490  the word not is repeated. 

Author Response

Madrid, 27th July 2022

Dear reviewer 2

We are pleased to submit for publication the revised version of the manuscript. At the following, the pints mentioned will be discussed:

1. line 86  ER2 should be ERBB2.

1. Thanks for this remark. We have changed it.

2. line 335 ....the paper of dual ERBB2.... the word paper should be efficacy I think. 

2. Thanks for this remark. We have changed it.

3. 'The MOUNTAINEER study was an open-label, single-arm phase was a phase 2 clinical trial when' ....line 355 needs to be rephrased.

3. Thanks for this remark. We have changed it.

4. line 490  the word not is repeated. 

4. Thanks for this remark. We have changed it.

Thank you for the revision. Kind regards.

Dr Javier Torres
